# Transcriptome Analysis Reveals the Molecular Mechanism and Responsive Genes of Waterlogging Stress in *Actinidia deliciosa* Planch Kiwifruit Plants

**DOI:** 10.3390/ijms242115887

**Published:** 2023-11-01

**Authors:** Mengyun Xing, Kangkang Huang, Chen Zhang, Dujun Xi, Huifeng Luo, Jiabo Pei, Ruoxin Ruan, Hui Liu

**Affiliations:** Hangzhou Academy of Agricultural Sciences, Hangzhou 310024, China; myxing@zju.edu.cn (M.X.); hkk34511@163.com (K.H.); tt.hang@163.com (C.Z.); xidujun@163.com (D.X.); huifengluo@163.com (H.L.); peijiabo@163.com (J.P.); buffalo126@126.com (R.R.)

**Keywords:** waterlogging, kiwifruit, ADH, RNA sequencing, stress

## Abstract

Waterlogging stress is one of the major natural issues resulting in stunted growth and loss of agricultural productivity. Cultivated kiwifruits are popular for their rich vitamin C content and unique flavor among consumers, while commonly sensitive to waterlogging stress. The wild kiwifruit plants are usually obliged to survive in harsh environments. Here, we carried out a transcriptome analysis by high-throughput RNA sequencing using the root tissues of *Actinidia deliciosa* (a wild resource with stress-tolerant phenotype) after waterlogging for 0 d, 3 d, and 7 d. Based on the RNA sequencing data, a high number of differentially expressed genes (DEGs) were identified in roots under waterlogging treatment, which were significantly enriched into four biological processes, including stress response, metabolic processes, molecular transport, and mitotic organization, by gene ontology (GO) simplify enrichment analysis. Among these DEGs, the hypoxia-related genes *AdADH1* and *AdADH2* were correlated well with the contents of acetaldehyde and ethanol, and three transcription factors Acc26216, Acc08443, and Acc16908 were highly correlated with both *AdADH1*/*2* genes and contents of acetaldehyde and ethanol. In addition, we found that there might be an evident difference among the promoter sequences of *ADH* genes from *A. deliciosa* and *A. chinensis*. Taken together, our results provide additional information on the waterlogging response in wild kiwifruit plants.

## 1. Introduction

Waterlogging stress, caused by continuous or excessive rain and poor soil drainage, is becoming one of the most common constraints for plant growth and development in the agricultural system. It was estimated that about 10% of the global land area was affected by waterlogging stress [1], and the yield production resulting from waterlogging to crops was reported up to 80% [2]. Conceivably, waterlogging stress is associated with potential economic afflictions. Statistically, the indirect economic loss from crop production caused by waterlogging was assessed to be the second largest one after drought [3]. With global climate change, the duration and intensity of extreme precipitation are increasing, and developing waterlogging tolerant varieties to maintain crop production is crucial for plant breeders.

The adverse impacts of waterlogging on plant crops mainly result from a low-oxygen environment. In waterlogged soil, root tissues are first submerged in water, and the concentrations of oxygen rapidly decline around the root rhizosphere, which is the main constraint for plant survival and growth. Hypoxia stress was regarded as an obstacle to aerobic respiration in roots, resulting in a lack of energy with root activity decreasing [4,5]. The increased concentrations of carbon dioxide around the roots are thought to enhance anaerobic metabolism, such as fermentation. During this process, a large number of toxic substances including lactic acid, ethanol, and acetaldehyde were produced [5,6]. Moreover, waterlogging stress influences the growth and development of plant crops by generating excessive reactive oxygen species (ROS), resulting in oxidative damage to cell tissues directly or indirectly. The root tissues were capable of responding to the adverse effects of hypoxia stress through their own antioxidant system by increasing the activities of antioxidant enzymes including superoxide dismutase (SOD), peroxidase (POD), and catalase (CAT) [7,8]. To survive in waterlogging stress, some root tissues trended to produce increased amounts of aerenchyma, providing an internal path for gas-phase diffusion with low-resistance of oxygen into and along roots [9]. In addition, the validity of available nutrients in the soil solution was changed by waterlogging, affecting the absorption, utilization, conversion, and redistribution of mineral nutrients in plant crops [10,11,12].

As a source of energy supply compounds, the accumulation of carbohydrates contributes to enhancing the tolerance to hypoxia stress. Numerous investigations have shown that the higher utilization of carbohydrates specifically sugars through the glycolytic pathway increases the survival rate during oxygen deficiency [13,14,15]. During hypoxia stress, sugar availability is one of the elements required for anaerobic metabolism maintaining the rapid reduction of cellular energy. In this process, the degradation of starch is needed to avoid sugar starvation leading to rapid cell death [4]. Under waterlogging stress, the anaerobic metabolism-related enzyme pyruvate decarboxylase (PDC) catalyzes pyruvate to produce acetaldehyde, which is metabolized by another protein alcohol dehydrogenase (ADH) to generate alcohol, with regeneration energy of NAD^+^ to sustain glycolysis for the growth of plant roots. The overexpressed and deficient genotypes of the *PDC* or *ADH* gene confirmed the essentiality of alcohol fermentation in the acclimation to waterlogging stress [16,17,18]. Therefore, glycolysis and alcohol fermentation are considered to alleviate the energy shortage caused by oxygen deficiency, which is of great significance in sustaining growth for plants under hypoxia stress.

Kiwifruit is popular for its rich vitamin C content and unique flavor among consumers. However, the majority of current kiwifruit cultivars are extremely sensitive to waterlogging stress because of their shallow and fleshy root system and high transpiration rate [19,20,21]. It is well known that the wild germplasm resources of kiwifruit plants are abundant. Nevertheless, the waterlogging adaptation strategy of these wild kiwifruit plants remains unknown. Previous studies have revealed information on the waterlogging tolerance of cultivated kiwifruit plats based on transcriptomic analysis [21,22]. Here, we carried out a comparative study on the molecular responses to waterlogging of wild kiwifruit germplasm (*Actinidia deliciosa*) using transcriptomic analysis. Our results provided evidence that the waterlogging tolerance of the wild germplasm marked Y20 from *A. deliciosa* was associated with metabolic adaptions with enhanced ethanolic fermentation, carbohydrate metabolism, and antioxidant abilities during waterlogging.

## 2. Results

### 2.1. Accumulation of Acetaldehyde and Ethanol in Kiwifruit Roots

The roots of kiwifruit showed no significant differences after waterlogged for three days compared with the control, while, seven days later the browning phenomenon occurred (Figure 1a). Acetaldehyde and ethanol are two main products of anaerobic respiration, which reflect the tolerance of plants to waterlogging stress. Our results exhibited that the endogenous acetaldehyde and ethanol accumulated significantly under waterlogging treatment (Figure 1b,c). In detail, the contents of acetaldehyde were increased to 66.74 μg/g fresh weight (FW) at three days after waterlogging (W3d) compared to the control (W0d), while reduced to 35.08 μg/g FW at W7d compared to the W3d although showed a significant increase compared to the W0d (Figure 1b). Accumulation of ethanol was induced significantly by waterlogging stress, and its changing pattern was similar to that of acetaldehyde. Actually, an evident increase to 93.10 μg/g FW at W3d and 52.80 μg/g FW at W7d of the ethanol content was detected in the root tissues (Figure 1c).

### 2.2. Analysis of the RNA Sequencing Data

The whole transcriptome sequencing analysis of the kiwifruit roots was performed with three biological replicates for each sample (W0dA, W0dB, W0dC, W3dA, W3dB, W3dC, W7dA, W7dB, W7dC). The paired-end sequencing with the Illumina NovaSeq6000 system was used to generate a total of 57.69 Gb clean data. To assess the quality of the obtained RNA-seq data, FastQC was used to assign a quality score (Q) of each base in the reads with a *phred*-like algorithm [23]. As a result, all clean read data had a *phred*-like quality score above Q20 (Q phred = 20) exceeded 97.30%, and quality scores above Q30 (Q phred = 30) exceeded 92.78% (Appendix A). After quality control, the paired-end reads with 45.92–47.06% of GC content (proportion of guanine and cytosine bases) were used in transcriptomic mapping analysis, and all groups were independently aligned with the reference Red5 genome using HISAT2 software. According to our results, about 79.47–82.74% of the reads were mapped to the kiwifruit genome (Appendix A). In addition, the mapped reads in each group were processed using the StringTie tool, and the assembled contigs for novel transcripts and genes were compared with the original genomic annotation information. As a result, a total of 36,235 transcripts were derived, among which 6449 may produce new genes (Appendix A).

Considering the fact that the annotation of the reference genome is usually not precise enough, it is necessary to align the assembled sequences to the Nr (non-redundant protein database), Pfam (protein families database), Swiss-Prot, eggNOG (a database of orthologous groups and functional annotation), GO (gene ontology knowledge base), and KEGG (Kyoto encyclopedia of genes and genomes) databases in the present study. Based on our results, there were 35,152 transcripts annotated by Nr databases, accounting for 97.01 percent of the total transcripts, and 30,778 ones by eggNOG (84.94%), 30,297 ones by GO (83.61%), 29,526 ones by Pfam (81.48%), 25,553 ones by Swiss-Prot (70.52%), and 10,525 ones by KEGG (29.05%) respectively (Figure 2a). In summary, these results confirmed that the obtained RNA-seq data were highly reliable, and they were suitable for subsequent analyses.

### 2.3. The Molecular Influence of Waterlogging on Root Tissues

During waterlogging stress, a series of endogenous genes were induced or inhibited to adapt to the outer harsh environment. In the present study, the fragments per kilobase of transcript per million fragments mapped (FPKM) values were normalized to calculate the gene expression levels. To assess the global differences between control and waterlogged groups, we carried out principal component analysis (PCA) and hierarchical clustering based on Pearson’s correlation coefficient of the average FPKM values for all 36,235 transcripts. As a result, the three biological replicates of control and waterlogged groups were clustered closely together of the first and second principal components respectively in a scatter plot (Figure 2b), indicating that our experimental processing was effective. Reasonably, the significant correlation among biological replicate samples was evidenced by high correlation coefficients exceeding 0.97 (Figure 2c), indicating that the three biological replicates in each waterlogged treatment own good reproducibility.

In order to demonstrate the reliability and adequacy of the waterlogged treatments, gene expression level distributions were calculated. The density plots of each sample library showed that the differences in the gene expression level distributions were low among the three repeated libraries of each group (Figure 2d), indicating the overall expression abundance of different samples was acceptable and the dispersion degree of expression distributions was average. Taken together, these results suggested that our waterlogged processing was effective, and the obtained sequencing data was reliable to mine the potential key genes involved in waterlogging stress.

### 2.4. Differential Gene Expression Analysis

To detect genes that were significantly differentially changed during waterlogging stress, DEG analysis was conducted by DESeq2. In our study, transcripts with Fold Change (FC) above 2 and False Discovery Rate (FDR) below 0.01 were defined as DEGs, which were identified by pairwise comparisons among the sample libraries. As a result, a total of 10,073 (5347 up- and 4726 down-regulated), 9380 (5096 up- and 4284 down-regulated), and 6072 (3078 up- and 2994 down-regulated) DEGs were identified in the analysis of W0d vs. W3d, W0d vs. W7d, and W3d vs. W7d, respectively (Figure 3). It was evident that the highest number of DEGs was detected in the comparison of W0d vs. W3d, while the fewest DEGs between W3d and W7d were counted (Figure 3a), indicating that most genes were changed at an early stage during waterlogging stress. Among all the DEGs, about 240 up-regulated and 106 down-regulated genes were commonly present in the three differential groups (Figure 3b,c), which indicated that the key genes related to waterlogging response might exist in the total 346 DEGs.

Considering the accumulation pattern of acetaldehyde and ethanol during waterlogging in the present study, we performed the specialized analysis of up- and down-regulated DEGs. The results showed that there were 283 genes up-regulated in W0d/W3d and W0d/W7d and down-regulated in W3d/W7d, which was consistent with the contents of acetaldehyde and ethanol (Figure 3d). In total, we obtained 629 candidate genes to explore the key ones related to waterlogging response.

### 2.5. GO Categories and Enrichment Analysis

Gene functional characterization and enrichment analysis are powerful tools for investigating DEGs to gain an understanding of their important functions and molecular pathways. Due to our transcripts being poorly annotated by the KEGG database, the GO enrichment was selected to evaluate the biological processes and molecular functions of the DEGs. Simplifying enrichment by clustering them into groups is a basic bioinformatic tool to explore the potential functions among the DEGs generated during waterlogging. Because the clusterings of the binary cut method were able to generate clean clusters and were capable of identifying large and small clusters at the same time [24], our 300 random GO terms were uniformly sampled by a binary cut from the biological processes ontology. Results showed that the DEGs were more significantly enriched in stress response, metabolic processes, molecular transport, and mitotic organization (Figure 4a).

In order to understand the molecular responding mechanism during waterlogging, a detailed GO enrichment analysis of DEGs was carried out. In total, there were 8139, 7570, and 4900 DEGs annotated by the GO database in the the comparisons of W0d vs. W3d, W0d vs. W7d, and W3d vs. W7d, respectively. With the *p*-value below 0.001 as the significance threshold, about 26, 36, and 37 GO classes were significantly enriched in different GO terms (Appendix A). Compared to the control, the GO enrichment analysis found that the significantly enriched DEGs in W0d vs. W3d were related to oxidoreductase activity, plasma membrane, UDP-glycosyltransferase activity, an integral component of membrane, and protein phosphatase inhibitor activity (Figure 4b). In the comparisons of W0d vs. W7d, the enriched DEGs were mainly associated with monooxygenase activity, oxidoreductase activity, protein phosphatase inhibitor activity, UDP-glycosyltransferase activity, protein kinase activity, and photosystem I (Figure 4c). In the analysis of W3d vs. W7d, about 7 pathways related to photosystem I, photosystem I reaction center, hydrolase activity, extracellular region, monooxygenase activity, oxidoreductase activity, and UDP-glycosyltransferase activity were significantly enriched (Figure 4d).

### 2.6. Identification of Candidate Genes under Waterlogging

Based on the GO enrichment analysis, about 24 DEGs associated with oxidoreductase activity and 17 ones related to UDP-glycosyltransferase activity were found up- or down-regulated all the time during waterlogging stress (Appendix A). To narrow down the screening scope of candidate genes, we set parameters with FPKM values above 50 and up-regulated log_2_FC above 3 or down-regulated log_2_FC below −2 as an acceptable threshold in the present study. In consequence, only Acc10216, which was down-regulated in W0d vs. W3d and W0d vs. W7d and up-regulated in W3d vs. W7d, was screened out according to our setting parameters (Appendix A). Following these screened rules, about 4 up-regulated and 24 down-regulated candidate genes were obtained among the 346 DEGs analyzed in the former part (Appendix A). Considering the fact that the prepared 283 DEGs were up- and down-regulated at different waterlogged stages in the comparisons of the three sample groups, we changed the parameters to up-regulated log_2_FC above 3 and down-regulated log_2_FC below −2 with FPKM values above 50. As a result, a total of 21 candidate genes were reserved from the 283 DEGs, among which only 50 transcripts were highly expressed with FPKM values above 100, and about 83 transcripts were accorded with the FPKM rules above 50 (Figure 5; Appendix A). Finally, we obtained 50 total candidate genes related to waterlogging response possibly.

Based on current knowledge of excessive water stress, we further analyzed the candidate 50 transcripts by their predictive functional annotations. Reasonably, the anaerobic respiration-related genes *AdADH1* and *AdADH2* were first selected as the key genes responding to waterlogging. Thus, we calculated the correlation relationship (Pearson *r*) of transcript levels between the seven transcription factors (TF) and *ADH* genes among the 50 transcripts. Results showed that there were three TFs named Acc26216, Acc08443, and Acc16908 correlated well with both *AdADH1* and *AdADH2* genes among the seven screened candidates (Figure 6a). Moreover, we further found that the expression levels of the three TFs and two *AdADH* genes showed a good correlation with the contents of both acetaldehyde and ethanol (Figure 1 and Figure 6), indicating these genes were possibly involved in the waterlogging response. Thus, the expression of the five candidate genes was deservedly verified by RT-qPCR (Figure 6b,c).

### 2.7. Analysis of Promoter Sequences of ADH Genes

To explore the regulation factors that responded to hypoxia stress, we carried out the project of promoter sequences separation of *ADH* genes from the wild Y20 roots. Considering the fact that none of the genome sequences of our wild kiwifruit were obtained, the genome data of *A. chinensis* ‘Red5’ and *A. deliciosa* was used as a reference to separate the *ADH* promoter sequences. As a result, promoters of both *AdADH1* and *AdADH2* could only be cloned with one primer pair separately among our designed primers, and the obtained sequences were quite short, indicating that there might be a great difference in the promoters between our wild Y20 and the reference *A. chinensis* and *A. deliciosa* (Table 1; Appendix A).

In addition, we analyzed the cis-elements in the promoters of *AdADH1* and *AdADH2* to see the potential transcription factors binding to these specific sequences. It is well known that the ERF, WRKY, MYB, and NAC transcription factors were involved in the hypoxia response, we focused on the binding sites of these four transcription factors in the *ADH* promoters. Results showed that ERF, MYB, and NAC had the potential to bind to both *AdADH1* and *AdADH2* promoters, while the WRKY could bind to the *AdADH1* promoter but was unable to bind to the obtained sequence of AdADH2 promoter (Figure 7a). We further found that promoters of both *AdADH1* and *AdADH2* contained the binding sites of GATA, MYB, ERF, Dof, bZIP, C2H2, NAC, and TBP transcription factors (Figure 7b). Variously, the *AdADH1* promoter included the binding sites of NF-YA/B/C, bHLH, TCR, and SBP transcription factors, and the *AdADH2* promoter contained the binding sites of AT-Hook, TALE, ZF-HD, and ARID (Figure 7b).

## 3. Discussion

Waterlogging stress negatively affects plant growth and development, and results in roots submerged in water with a rapid decline in the oxygen concentration around the root rhizosphere. Under hypoxia condition, aerobic respiration in roots is limited causing a result of insufficient energy, which induces anaerobic respiration, metabolites reduction, cytoplasmic acidosis, oxidative stress, toxic chemicals accumulation, nutrient deficiency, and root injury or even plant death [25,26,27]. The abundant wild germplasm resources and rich genetic diversity of kiwifruit plants are natural resources for improving the waterlogging tolerance of kiwifruit varieties. In our present results, the wild *A. deliciosa* germplasm displayed slight symptoms of an injury without evident rotten roots under waterlogging treatment (Figure 1), indicating that this wild Y20 germplasm has the potential to resist waterlogging stress and be a promising genetic candidate resource for breeding waterlogging-tolerant kiwifruit rootstocks or cultivars.

Considering the fact that transcriptome sequencing technology has been widely applied to elucidate the molecular response mechanisms of waterlogging resistance, we conducted this technology to study the transcriptional response of waterlogging stresses in kiwifruit plants. Results showed that the obtained sequencing data in the present study was reliable and our waterlogged treatment of kiwifruit roots was effective (Figure 2). Previous studies have provided information on variations in the number of responded transcripts of kiwifruit genotypes with different tolerance levels under oxygen deprivation stress [28,29,30]. In this study, we found that there were more up-regulated DEGs than down-regulated ones (Figure 3), indicating the expression induction and enhancement of gene function were more responsible in response to waterlogging stress of *A. deliciosa* roots, consistent with previous study in *A. valvata* [30]. Compared with roots soaked in water for 3 days, the number of DEGs decreased from 10,073 to 6072 in the roots waterlogged in water for 7 days (Figure 3), suggesting that the waterlogging increased stress in the kiwifruit roots over time along with drastic changes occurring at 3 days. Similar response trends of waterlogging stress have been reported in waterlogged *Arabidopsis* roots [31], *Cerasus sachalinensis* roots [32], and *Dactylis glomerata* leaves [33]. Thus, waterlogging tolerance is a complicated trait with multifarious changes on the molecular levels.

With long-term evolution, plants have developed highly complex mechanisms to cope with intricate environmental stress [34,35,36,37]. In the present study, the biological metabolisms including stress response, metabolic processes, molecular transport, and mitotic organization were the main physiological processes of the wild kiwifruit responding to waterlogging stress (Figure 4). Under saturated water stress, plants are forced to produce excessive reactive oxygen species, which could cause progressive oxidative damage to the cell membranes [38,39]. To alleviate this injury, plants always activate their own antioxidant defense systems to scavenge the excessive reactive oxygen species, including SOD, CAT, POD, and glutathione S-transferase (GST) [6,7,40]. In addition, energy metabolism is essential for plant survival under waterlogging stress, which relies on the activation of starch and sucrose metabolism and the glycolysis process [41,42,43]. Our results showed that transcripts in the wild kiwifruit related to cell membrane system, oxidoreductase activity, and carbohydrate metabolism were significantly regulated during the waterlogging response (Figure 4), consistent with previous studies. Taken together, the *A. deliciosa* kiwifruit dealt with waterlogging damage mainly by activating its own antioxidant defense systems and fermentation metabolism in the present study.

Waterlogging induces numerous physiological and metabolic processes, among which anaerobic respiration is the primary one for the energy production of ATP [43,44]. In current knowledge, ethanolic fermentation as the main form of anaerobic respiration has been classically associated with waterlogging tolerance. During this process, the ADH oxidizing ethanol to acetaldehyde is one of the key enzymes and is regarded as a molecular marker for reflecting the stress response to hypoxia. As a result, activation in alcohol fermentation catalyzed by ADH has been repeatedly reported in different plant crops during waterlogging response [22,45]. For example, it was reported that the expression levels of *AdADH1* and *AdADH2* were significantly increased based on Illumina sequencing technology in *A. deliciosa* after treatment with waterlogging, and transgenic overexpressing *AdADH1* or *AdADH2* was able to enhance waterlogging tolerance of Arabidopsis seedlings [18]. In our results, the *AdADH1* and *AdADH2* genes were significantly up-regulated under waterlogging treatment (Figure 5 and Figure 6), which helps to provide small amounts of NAD^+^ to maintain glycolysis for root growth. Hence, changes in gene expression related to alcohol fermentation provided evidence to understand how the wild A. *deliciosa* endures lengthy waterlogging stress, and its molecular mechanisms remain to be further studied.

To survive under adverse stress, plants always change their own gene expression and regulate different metabolic processes or signal transduction pathways at different levels. As we all know, transcription factors play crucial roles in the regulation of various abiotic stress responses. Numerous investigations have shown that ERF-VII transcription factors were the master players in the regulation of tolerance responses to waterlogging stress by enhancing hypoxia-related ADH activity [46,47]. Subsequently, additional TFs, including WRKY, NAC, and MYB were proposed to be involved in waterlogging responses [48,49,50]. In our results, three transcription factors Acc26216, Acc08443, and Acc16908 were significantly induced under the waterlogging stress and were highly correlated with both *AdADH1*/*2* genes and contents of acetaldehyde and ethanol, indicating a relationship between the three transcription factors and the two *ADH* genes (Figure 6). Thus, we tried to separate the promoters of *AdADH1* and *AdADH2* from the wild Y20 roots to search for the potential transcription factors responding to waterlogging stress. It was a pity that the promoter sequences were difficult to separate, and we only obtained the 200–300 bp upstream fragments of the starting codon of the *ADH* genes (Table 1). Based on these above findings, we analyzed the binding sites in the obtained *ADH* promoters. Results showed there were no binding sites of the three candidate transcription factors existed in the obtained *AdADH1* and *AdADH2* promoters though the binding sites of ERF, WRKY, MYB, and NAC were contained in the promoter sequences (Figure 7). Previously, Tang et al. [51] proved that the Arabidopsis ERF-VII member RAP2.2 acted downstream of WRKY33 or WRKY12 by binding to and activating RAP2.2 individually during the submergence process, indicating a regulatory system formed by transcription factors mediating the hypoxia response. Taken together, the candidates Acc26216, Acc08443, and Acc16908 might be involved in waterlogging response by binding to the unobtained promoter sequences directly or interacting with other transcription factors indirectly.

## 4. Materials and Methods

### 4.1. Plant Materials and Waterlogging Stress Treatment

The experiment was conducted in a greenhouse with 75% relative humidity and 25 °C air temperature. The light intensity was approximately 100 mmol^−2^ in 16 h light/8 h dark long-day conditions. Seeds of our wild kiwifruit (*Actinidia deliciosa* Planch) obtained from Daqi mountain in Zhejiang province of China were sown in soppy peat-based growing media. Two months later, plantlets with five mature leaves were transferred to 10-cm-diameter pots containing a mixture of peat moss, perlite, vermiculite, and rice hull. The waterlogging stress treatment was performed when vigorous seedlings owned about ten leaves. According to Liu et al. [21], two groups of potted plants were placed in a plastic container, the treated group was filled with tap water above the soil surface 4–5 cm and the control one was cultured with normal irrigation. Root tissues were collected after 0, 3, and 7 d of irradiation from three individual plants, and freshly harvested materials were rapidly frozen in liquid nitrogen and stored at −80 °C condition for further studies.

### 4.2. Acetaldehyde and Ethanol Contents

The root samples of kiwifruit were ground into a fine powder with a sample grinder while frozen. According to Liu et al. [34], extraction, detection, and quantification of acetaldehyde and ethanol were carried out with a minor modification. About 200 mg root powders were fully mixed in 2 mL saturated NaCl buffer, and the supernatants were harvested with centrifugation for 15 min at 12,000 rpm. Transferring 1.5 mL of the supernatants into a headspace extraction bottle with 0.2% 2-butanol as an inner reference, the mixtures were then incubated at 60 °C in a water bath for 1 h. Separation of purified extracts was performed using an Agilent 6890N gas chromatograph (GC, Palo Alto, CA, USA), which was equipped with an HP-INNOWAX capillary column, according to the methods described by Liu et al. [21]. Pure nitrogen was used as a carrier gas, and the temperatures of the oven, back-inlet, and front-detector were set as 100 °C, 150 °C, and 160 °C, respectively. Contents of acetaldehyde and ethanol chemicals were calculated with their own standard curves according to the chromatographic peak areas in the Agilent gas chromatograph system.

### 4.3. RNA Extraction and Library Preparation

Total RNA was extracted from kiwifruit root samples using the CTAB method according to Wang et al. [52], and treated with DNase I to remove genomic DNA. The concentration and purity of extracted RNA were determined using the NanoDrop2000 spectrophotometer, and quality was determined using 1% denaturing agarose gels. RNA integrity was assessed using the Agient2100/LabChip GX bioanalyzer (Agilent Technologies, Santa Clara, CA, USA). For sequencing library construction, enrichment of mRNA from total RNA was carried out using oligo (dT) magnetic beads. The cDNA was synthesized using mRNA fragments as templates, and the fragments of double-stranded cDNAs were filtered using the AMPure XP beads. The quality of cDNA libraries was tested using a Qsep400 high-throughput analysis system, and ensured by quantitative PCR.

### 4.4. RNA Sequencing and Assembly

The RNA sequencing was performed by staff at Biomarker Technologies Co., Ltd. (Beijing, China) on an Illumina NovaSeq6000 platform with pair-end 150 nucleotides. Raw data of RNA-Seq was filtered by removing reads containing adapters, reads containing ploy-*N* > 10%, and low-quality (Q-value  ≤  10) bases. Simultaneously, the Q20, Q30, GC content and sequence duplication levels of the clean data were calculated. All the following analyses were based on clean data with high quality. As described by Wang et al. [53], the ‘Red5’ kiwifruit genome was built as reference information [54]. Thus, the high-quality clean reads were mapped to the reference genome using the HISAT2 tool and subsequently assembled using the StringTie software. Unique transcripts were functionally annotated according to NR, Swiss-Prot, COG, KOG, GO, and KEGG databases using DIAMOND software, and based on the Pfam database using HMMER software.

### 4.5. Differential Gene Expression Analysis

Fragments per kilobase of transcript per million fragments mapped (FPKM) values of each gene were used to calculate the expression levels based on StringTie software. Principal component analysis (PCA) and correlation coefficient were conducted to determine the relatedness of the biological replicates. The DESeq2 tool was used to perform differential expression (DEG) analysis by pairwise comparisons among the sample libraries. The false discovery rate (FDR) was considered to determine the threshold of the resulting *p*-values in multiple tests, and an FDR below 0.01 and fold change above 2 were considered as the cutoff thresholds to determine the significance of gene expression. For functional enrichment analysis, significantly up- and down-regulated genes were selected for GO analysis by using the R package or the online PlantPAN4.0 website.

### 4.6. RT-qPCR Assays

According to the manufacturer’s instructions, our prepared RNA was reverse transcribed into cDNA with PrimeScript™ RT reagent Kit (Takara, Kusatsu, Japan). Real-time fluorescence quantitative test was performed on a Roche LightCycler^®^ 480 device according to the procedure described by Wang et al. [52] using Roche LightCycler^®^ 480 SYBR Green I Master, and actin was selected as the internal reference gene to normalize the expression levels. All gene-specific primers checked by PCR re-sequencing and melting-curve are shown in Appendix A. The relative expression levels of kiwifruit genes were calculated by the 2^−∆CT^ method. Each reaction was repeated at least three times.

### 4.7. Statistical Analyses

Figures were produced with GraphPad Prism 7 (GraphPad Software Inc., Boston, MA, USA). The two-tailed Student’s *t*-test was calculated by SPSS software (SPSS 19.0, SPSS Inc., Armonk, NY, USA). All data were obtained with at least three replicates, and error bars indicate standard error (SE).

## 5. Conclusions

Here, we analyzed the changes in the transcription levels of the wild *A. deliciosa* under waterlogging stress based on the RNA sequencing technology. Our results suggested that the wild Y20 was indeed capable of strongly responding to waterlogging-associated hypoxia at the transcriptional level in the root tissues. The metabolic adjustments with enhanced carbohydrate metabolism, ethanolic fermentation, and antioxidant abilities might contributed to the waterlogging adaption of the wild *A. deliciosa*. Under waterlogging stress, expression levels of *AdADH1* and *AdADH2* involved in ethanolic fermentation were significantly increased, which were positively consistent with the changes in acetaldehyde and ethanol contents. In addition, we screened out three candidate transcription factors Acc26216, Acc08443, and Acc16908, whose expression showed good correlations with expressions of *ADH* genes and contents of acetaldehyde and ethanol. However, there were no binding sites of these three transcription factors in the obtained promoters of *AdADH1* and *AdADH2*, indicating more studies focusing on the relationship of *ADH* genes and three candidate factors remained to be carried out.

## Figures and Tables

**Figure 1 ijms-24-15887-f001:**
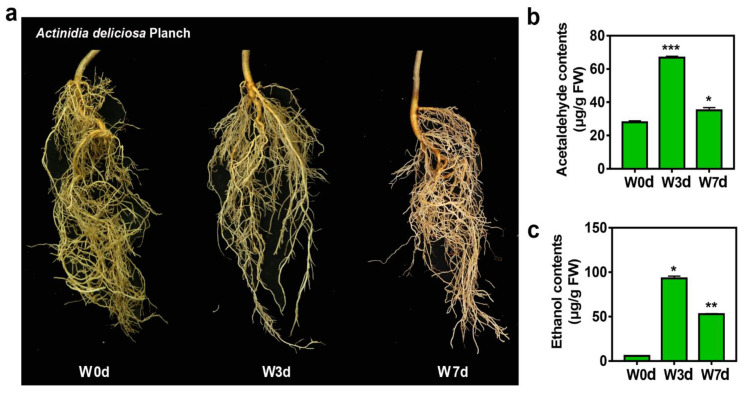
Morphological changes of kiwifruit roots under waterlogging stress. (**a**) Phenotypic differences of kiwifruit roots after waterlogged with 0 days (W0d), 3 days (W3d), and 7 days (W7d). (**b**) The acetaldehyde content in kiwifruit roots. (**c**) The ethanol content in kiwifruit roots. Error bars indicate ± S.E. from three biological replicates (* *p* < 0.05, ** *p* < 0.01, *** *p* < 0.001).

**Figure 2 ijms-24-15887-f002:**
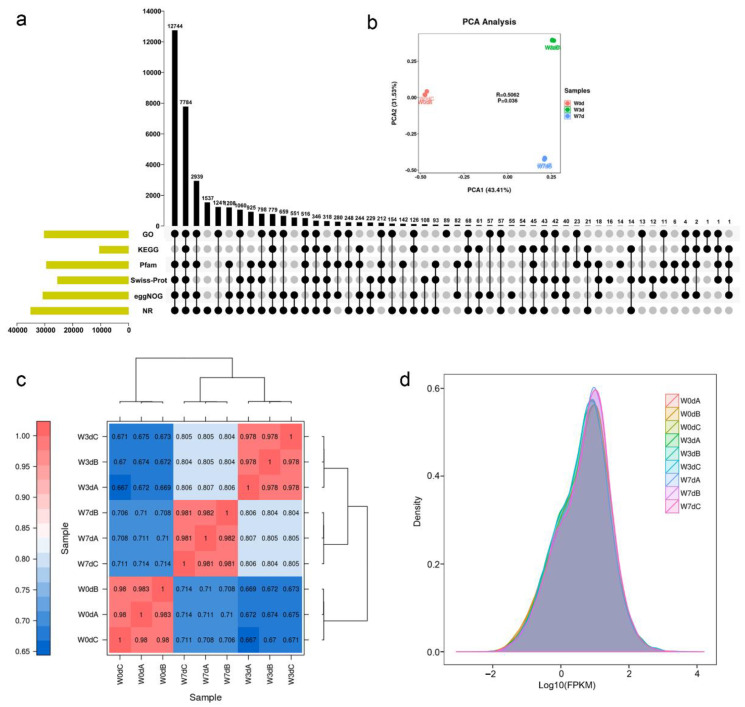
The analysis of RNA-seq in kiwifruit plants waterlogged with 0 days (W0d), 3 days (W3d), and 7 days (W7d). (**a**) Upset plot of transcript numbers annotated by Nr, eggNOG, Swiss-Plot, Pfam, KEGG, and GO databases. (**b**) PCA analysis of the three treatment groups. (**c**) Correlation analysis of the expression levels. (**d**) Density plot of expression distribution with logarithmic values of FPKM on the horizontal axis and density values on the vertical axis.

**Figure 3 ijms-24-15887-f003:**
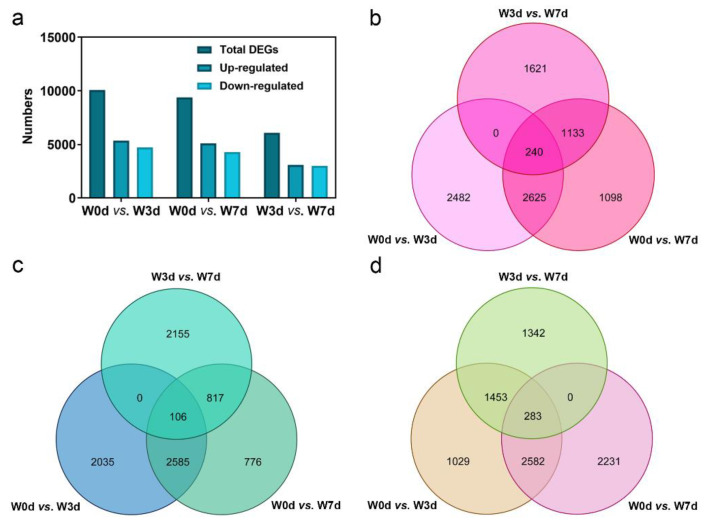
Identification of differentially expressed genes (DEGs) during waterlogging stress. (**a**) Numbers of DEGs in the three analyzed samples. (**b**) Venn diagram of up-regulated genes of all DEGs in each sample. (**c**) Venn diagram of down-regulated genes of all DEGs in each sample. (**d**) Comparison of up-regulated genes in W0d/W3d and W0d/W7d, and down-regulated genes in W3d/W7d.

**Figure 4 ijms-24-15887-f004:**
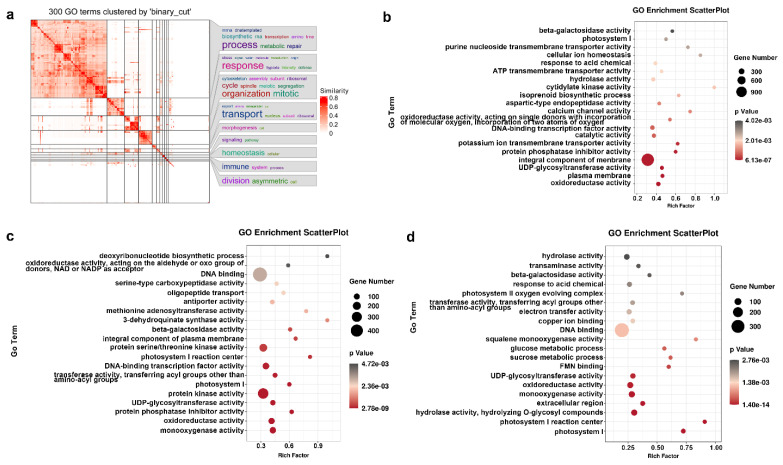
The Gene Ontology (GO) classification of differentially expressed genes (DEGs) in kiwifruit. (**a**) Similarity heatmap from 300 random GO terms clustered by the binary cut of all DEGs. The plot was made by the function *simplifyGO()*. (**b**) Go enrichment of DEGs between W0d and W3d. (**c**) Go enrichment of DEGs between W0d and W7d. (**d**) Go enrichment of DEGs between W3d and W7d. The color indicates terms with the *p*-values. Bubble size represents the ratio of DEG numbers to the total numbers of DEGs enriched on the GO terms.

**Figure 5 ijms-24-15887-f005:**
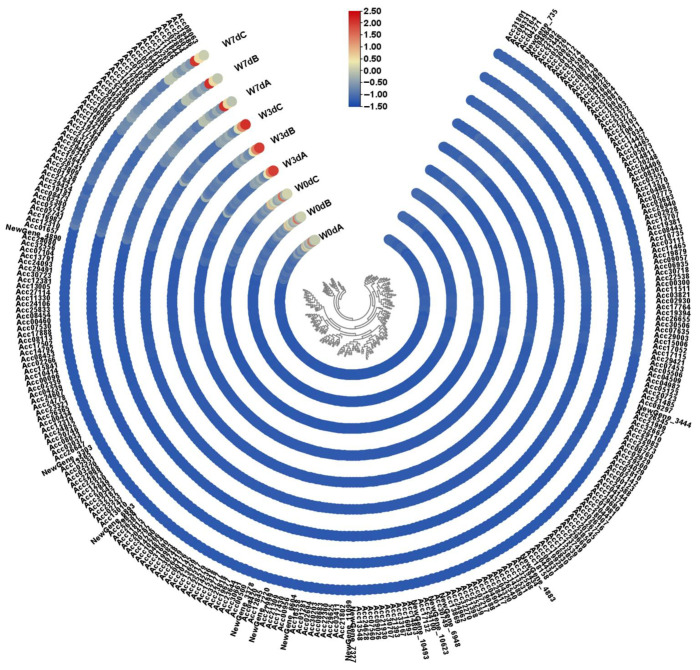
Expression pattern of the 283 genes in kiwifruit roots. The expression pattern was constructed based on the FPKM with log2 transformation and analyzed by heatmap hierarchical clustering. The color scale (representing −1.5 to 2.5) was shown.

**Figure 6 ijms-24-15887-f006:**
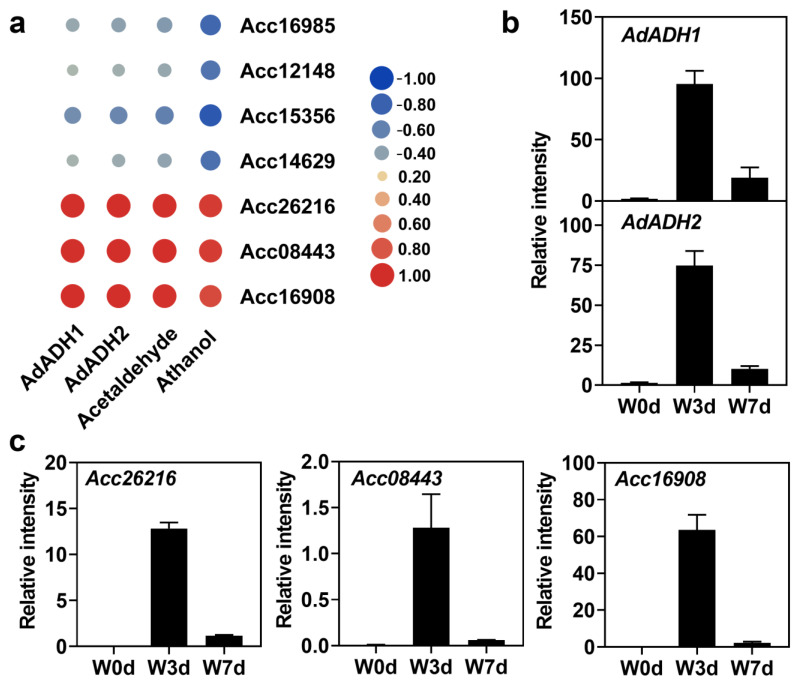
Identification of candidate genes under waterlogging stress. (**a**) Correlation analysis of transcript levels between candidate genes and acetaldehyde or ethanol content. (**b**) Expression profiles of *AdADH1* and *AdADH2* genes. (**c**) Expression profiles of the three candidate transcription factors verified by RT-qPCR.

**Figure 7 ijms-24-15887-f007:**
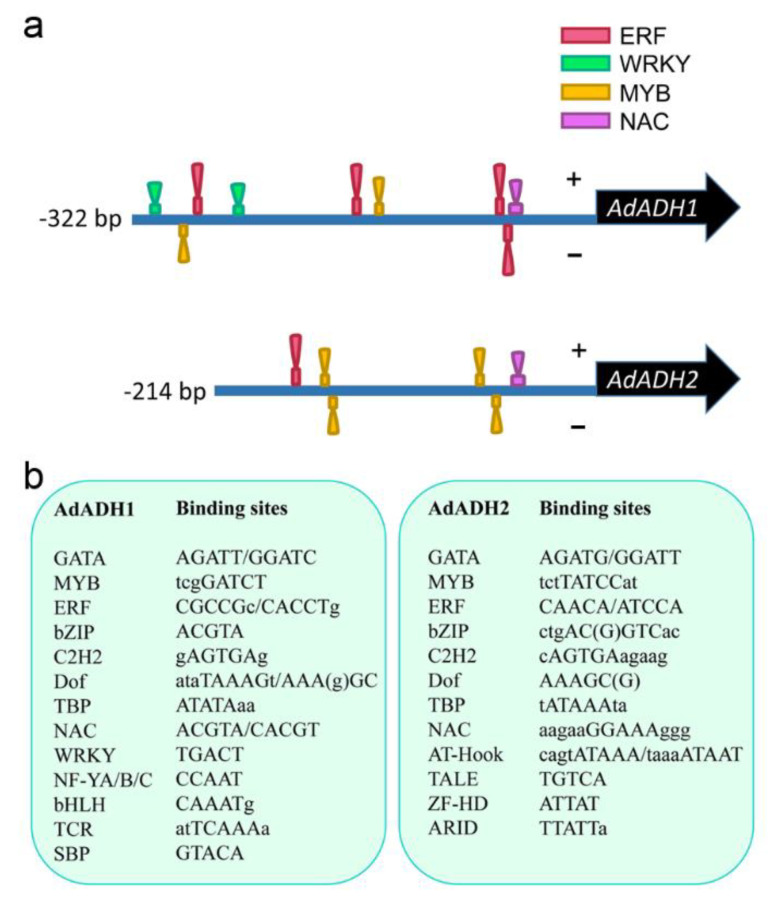
Analysis of the promoter sequences of *AdADH1* and *AdADH2* genes from the wild *Actinidia deliciosa*. (**a**) Schematic location of the binding sites in the *AdADH1* and *AdADH2* promoters for the potential transcription factors. (**b**) Information of the potential transcription factors binding sites in the *AdADH1* and *AdADH2* promoters with similarity score above 0.95. The promoter sequences were analyzed based on the PlantPAN4.0 website.

**Table 1 ijms-24-15887-t001:** Combination information of primers used for promoter sequences cloning of *AdADH1* and *AdADH2*. The word ‘No’ indicates the promoter sequences failed to be separated.

Gene.	Combinations	Reference Genome	Predicted Length (bp)	Obtained Length (bp)
*AdADH1*	FP1 + RP1 (FP1 + RP2)	‘Hayward’	2000 (1970)	No
	FP2 + RP1 (FP1 + RP2)	‘Red5’	2000 (1970)	No
	FP3 + RP1 (FP3 + RP2)	‘Hayward’	1342 (1312)	No
	FP4 + RP1 (FP4 + RP2)	‘Hayward’/‘Red5’	422/392	No
	FP5 + RP1 (FP5 + RP2)	‘Hayward’/‘Red5’	346 (316)/351 (321)	322 (No)
*AdADH2*	FP1 + RP1 (FP1 + RP2)	‘Hayward’	2000 (1976)	No
	FP2 + RP1 (FP1 + RP2)	‘Red5’	2000 (1976)	No
	FP3 + RP1 (FP3 + RP2)	‘Hayward’/‘Red5’	1556 (1532)/1295 (1271)	No
	FP4 + RP1 (FP4 + RP2)	‘Hayward’/‘Red5’	1140 (1116)/862 (838)	No
	FP5 + RP1 (FP5 + RP2)	‘Hayward’/‘Red5’	864 (840)/586 (562)	No
	FP6 + RP1 (FP6 + RP2)	‘Hayward’/‘Red5’	645 (621)/379 (355)	No
	FP7 + RP1 (FP7 + RP2)	‘Hayward’/‘Red5’	235 (211)/230 (206)	214 (No)

## Data Availability

The transcriptome sequencing data in present study was available in the NCBI/SRA database (BioProject accession number: PRJNA1033716).

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
