# Peer review of "Transcriptome Analysis Reveals the Molecular Mechanism and Responsive Genes of Waterlogging Stress in Actinidia deliciosa Planch Kiwifruit Plants"

_ijms, 2023, doi:10.3390/ijms242115887_

Round 1

Reviewer 1 Report

Dear Authors,

The manuscript by Xing et al. investigates the impact of water stress on the root system of wild kiwi plants and their associated molecular response mechanisms. The research demonstrates that wild kiwifruit is capable of responding strongly to oxygen deficiency caused by waterlogging in root tissues. Additionally, the results suggest that plants adapt to waterlogging by adjusting their metabolism, such as increasing carbohydrate metabolism, ethanol fermentation, and antioxidant capacities.

Overall, the manuscript has scientific potential, but it requires some structural and content improvements to fully meet the expectations of a scientific publication and better articulate the significance and conclusions of the research.

Key critical observations:

  1. While the text provides detailed descriptions of the experimental methods, it lacks details regarding the results and their significance. The summary and concluding sections of the results are missing.
  2. The evaluation of the results is brief and does not provide deeper analysis or practical implications.
  3. Further information is needed to understand the complexity of the research regarding the relationship of transcription factors without binding sites in ADH gene promoters.

Detailed critical comments:

Abstract: The abstract is concise and clear but does not mention the main hypothesis or objectives of the research. It fails to highlight the critical findings and their significance, making it challenging for readers to grasp the research's importance.

Introduction: I think this is the most problematic section. While it discusses the effects of water stress on plants, it does not explain why this research is essential and interesting to readers. The research question or hypothesis that guided the study, as mentioned in the abstract, is missing.

Results: In this section Authors focuse on presenting the findings but lacks analysis of the underlying mechanisms or their significance. The summary and comparison of the results with the hypothesis or previous research are missing.

The Discussion section provides detailed descriptions of methods and results but places less emphasis on their significance or the research's conclusions. It lacks the presentation of disputed points or factors challenging the results.

Materials and Methods: This section thoroughly describes the experimental methods but fails to explain why these methods were chosen and how they were relevant to answering the research question. Some details, such as the timing of sampling or sample storage methods, are missing.

Conclusion: This section summarizes the main findings but does not mention the broader context of the research or its potential for further studies. A strong closing thought or summary sentence at the end of the text is missing.

Author Response

Dear editor and reviewer 1:

Thank you for your letter and reviews regarding our manuscript ijms-2648385. We have revised our manuscript according to your comments. The following are our response to the questions raised by reviewer 1:

Key critical observations:

Q1: While the text provides detailed descriptions of the experimental methods, it lacks details regarding the results and their significance. The summary and concluding sections of the results are missing.

Reply: In the results part of our manuscript, summary and concluding descriptions were listed in Line 89-90, 128-130, 155-157, and 182-183 of the manuscript.

Q2: The evaluation of the results is brief and does not provide deeper analysis or practical implications.

Reply: Yes, we agree with you that our present results are easy to understand. As we all know that deeper analysis always need more experimental data, and our present data is not enough to support more scientific information. But, we would think more about the present data and the research methods of waterlogging response according to your suggestions.

Q3: Further information is needed to understand the complexity of the research regarding the relationship of transcription factors without binding sites in ADH gene promoters.

Reply: Yes, we agree with you. The question is, there is no genome data of our wild kiwifruit, and  we fail to obtain more length of promoter sequences according to the reference genome. In fact, we are trying to separate more promoter sequences of ADH and PDC genes in our wild kiwifruit by molecular experiment, and we plan to do more jobs about the relationships between the candidate transcription factors and ADH/PDC genes.

Detailed critical comments:

Q4: Abstract: The abstract is concise and clear but does not mention the main hypothesis or objectives of the research. It fails to highlight the critical findings and their significance, making it challenging for readers to grasp the research's importance.

Reply: In the abstract, we emphasize the tolerance differences of waterlogging between cultivated and wild kiwifruits, and summarize the main findings. In this study, we analyze the transcriptome data, and confirm the molecular response of waterlogging in wild kiwifruit, which is consistent with previous study. While, it is hard to ensure the detail response process according to the present analysis. Hence, there is no hypothesis in the manuscript.

Q5: Introduction: I think this is the most problematic section. While it discusses the effects of water stress on plants, it does not explain why this research is essential and interesting to readers. The research question or hypothesis that guided the study, as mentioned in the abstract, is missing.

Reply: In Line 28-36, we introduce the adverse damages on agricultural crops, and emphasize the economic loss caused by waterlogging. We think, these are the requirement for crops and attraction to readers. In Line 72-78, we introduce the cultivated and wild kiwifruits, which is mentioned in the abstract, not missing.

Q6: Results: In this section Authors focuse on presenting the findings but lacks analysis of the underlying mechanisms or their significance. The summary and comparison of the results with the hypothesis or previous research are missing.

Reply: As replied above Q1, we summarize our main results in corresponding paragraph, not missing. In addition, the comparison of our results with previous research is carried out in detail in the discussion part other than results section.

Q7: The Discussion section provides detailed descriptions of methods and results but places less emphasis on their significance or the research's conclusions. It lacks the presentation of disputed points or factors challenging the results.

Reply: In the manuscript, we emphasize our findings and conclusion in the discussion part, such as Line 296-298, 314-315, and 329-331. The results of transcriptome data showed no disputed points compared with previous study in kiwifruit, and present data is insufficient to discuss the disputed points in detail compared with molecular findings in other plants.

Q8: Materials and Methods: This section thoroughly describes the experimental methods but fails to explain why these methods were chosen and how they were relevant to answering the research question. Some details, such as the timing of sampling or sample storage methods, are missing.

Reply: This is not a method research article, all the methods we used in present study are common molecular biology techniques, which were ensured by previous researchers. The timing and condition of sampling and storage were introduced in detail in Line 385-387, not missing.

Q9: Conclusion: This section summarizes the main findings but does not mention the broader context of the research or its potential for further studies. A strong closing thought or summary sentence at the end of the text is missing.

Reply: We summarize the study just based on our current findings, and at the end of this section, we introduced the research target in the future. Indeed, it is quite hard to write down a strong (what the ‘strong’ mean?) summary at the end of this text.

Sincerely yours

Hui Liu

Hangzhou Academy of Agricultural Sciences, Hangzhou 310024, PR China

Reviewer 2 Report

Dear Authors,

The manuscript is well-written. Please refer my comments for suggestions.

Can you discuss the role of downregulated genes that are relevant to hypoxia tolerance in the wild germplasm Y20? 

It is well-written. However, use scientific words.

Author Response

Dear reviewer 2:

Thanks for your reviews regarding our manuscript ijms-2648385. We have revised our manuscript according to your comments and kind suggestions. The following are our response to the questions raised by reviewer 2:

The manuscript is well-written. Please refer my comments for suggestions.

Q1: Can you discuss the role of down-regulated genes that are relevant to hypoxia tolerance in the wild germplasm Y20?

Reply: Thanks for your suggestions. Before draft the manuscript, we focused on all the changed genes, including the down-regulated ones. It seems that the down-regulated genes selected by our analytical methods, were not related to the alcohol fermentation although they showed well correlation with ethanol content (Figure 6a). Among the four down-regulated transcription factors Acc16985, Acc12148, Acc15356, and Acc14629, the expression of zinc transporter 1 Acc15356 showed better but low correlation with acetaldehyde content and expressions of AdADH1/2 genes, compared with other three ones. Considering the data of fold change (FC), the four down-regulated genes may participate in waterlogging response by unknown biological processes other alcohol fermentation. According to our knowledge, the damaged phenotype during waterlogging stress such as root vitality might be more helpful for identification of down-regulated genes .

Reply: According to your kind suggestions, we revised the manuscript focused on your annotations and marks. In the revised text, we retained two words the might in Line 21 and the adaptions in Line 82. Our sequence separation results indicate the potential differences among the promoter sequences of ADH genes, but the evidence is insufficient to support this conclusion, and the whole genome sequences of the ADH genes need to be obtained to verify this difference. We confirmed the description of metabolic adaptions is our means.

Q3: It is well-written. However, use scientific words.

Reply: Thanks for your suggestions, we corrected the corresponding words according to you.

Sincerely yours

Hui Liu

Hangzhou Academy of Agricultural Sciences, Hangzhou 310024, PR China

Round 2

Reviewer 1 Report

Dera Authors,

I consider the manuscript to be acceptable in its current state, and recommend it for publication.

Best Regards,

Reviewer

Dera szerzők,

A kéziratot jelenlegi állapotában elfogadhatónak tartom, közlésre ajánlom.

Üdvözlettel,

Bíráló

Reviewer 2 Report

Dear authors,

Thank you for making correction. As you pointed out that down regulated genes were associated with Zinc transporter. The possible reason could be a leakage of nutrients. 

All the Best!